



# The ionospheric response over the UK to major bombing raids during World War II

**Christopher J. Scott**[1] **and Patrick Major**[2]

[1]Department of Meteorology, University of Reading, Berkshire, UK
[2]Department of History, University of Reading, Berkshire, UK

**Correspondence:** Christopher J. Scott (chris.scott@reading.ac.uk)

Received: 11 May 2018 – Discussion started: 18 May 2018
Accepted: 14 August 2018 – Published:

**Abstract.** The Earth's ionosphere is subject to disturbance from above (via solar variability and space-weather effects) and from below (such as tectonic activity, thunderstorms and sudden stratospheric warmings). Identifying the relative contribution of these effects remains challenging, despite recent advances in spacecraft monitoring near-Earth space. Manmade explosions provide a quantifiable proxy for natural terrestrial sources, enabling their impact on ionospheric variability to be studied. In this paper, the contribution of ground-based disturbances to ionospheric variability is investigated by considering the response of the ionospheric F2 layer over Slough, UK, to 152 major bombing raids over Europe during World War II, using a superposed epoch analysis. The median response of the F2 layer is a significant decrease in peak electron concentration ( $\sim 0.3$ MHz decrease in $fo$F2). This response is consistent with wave energy heating the thermosphere, enhancing the (temperature-dependent) loss rate of $O^+$ ions. The analysis was repeated for a range of thresholds in both time of bombing before the (noon) ionospheric measurement and tonnage of bombs dropped per raid. It was found that significant ( $\sim 2$–$3\sigma$) deviations from the mean occurred for events occurring between approximately 3 and 7 h ahead of the noon ionospheric measurements and for raids using a minimum of between 100 and 800 t of high explosives. The most significant ionospheric response ($2.99\sigma$) occurred for 20 raids up to 5 h before the ionospheric measurement, each with a minimum of 300 t of explosives. To ensure that the observed ionospheric response cannot be attributable to space-weather sources, the analysis was restricted to those events for which the geomagnetic Ap index was less than 48 (Kp < 5). Digitisation of the early ionospheric data would enable the investigation into the response of additional ionospheric parameters (sporadic E, E and F1 layer heights and peak concentrations). One metric ton of TNT has an explosive energy of $4.184 \times 10^9$ J, which is of the same order of energy as a cloud to ground lightning stroke. Since the occurrence of lightning has distinctive diurnal and seasonal cycles, it is feasible that a similar mechanism could contribute to the observed seasonal anomaly in ionospheric F-region electron concentrations. Further investigation, using less extreme examples, is required to determine the minimum explosive energy required to generate a detectable ionospheric response.

## 1 Introduction

The source of much of the observed variability within the Earth's ionosphere remains poorly understood. In this study we examine unique ionospheric measurements made above Slough, UK, for the duration of the Second World War (WWII) in order to determine whether any of the observed variability could be attributed to the major bombing campaigns across Europe.

Production of ionisation in the Earth's upper atmosphere is predominantly through photo-ionisation by solar extreme ultra violet (EUV) and X-ray radiation, while loss rates are very sensitive to the temperature and composition of the neutral thermosphere. As a result, the long-term average behaviour of the ionosphere is closely tied to solar activity and is well understood. Transient space-weather phenomena such as coronal mass ejections, high-speed solar wind streams and energetic particle events can temporarily perturb the ambient ionospheric conditions by enhancing ionospheric production through impulsive brightening of the solar atmosphere (so-

lar flares), enhancing ionospheric loss rates (through heating of the thermosphere, affecting neutral composition and loss rates), and through direct enhancement of ionisation by precipitation of energetic particles. While the details of such processes are still the subject of ongoing research, once again, the underlying physics is broadly understood. Despite this, there are further sources of ionospheric variability that remain unaccounted for and it has been suggested (e.g. Rishbeth and Müller-Wodarg, 2006) that the source of this variability is from the lower atmosphere. Sources such as earthquakes (e.g. Astafyeva et al., 2013, and references therein), thunderstorms (Davis and Johnston, 2005; Yu et al., 2015) and explosions (e.g. Rishbeth, 1991) have been cited as potential causes of ionospheric variability, with a variety of proposed mechanisms including pressure waves, gravity waves, infrasound and modulation of the global electric circuit.

There have been a number of case studies into the impact of terrestrial explosions on the upper atmosphere (e.g. Rishbeth, 1991), most notably the events surrounding the explosion at the Flixborough chemical plant in 1974 (Jones and Spracklen, 1974; Krasnova et al., 2003), while Pohotelov et al. (1991) considered the impact on the F2 and Es ionospheric layers during a 32-day period of bombing in the Gulf War. In this current paper we make use of historical records to identify large bombing raids over mainland Europe during WWII and, using a superposed epoch, or composite analysis (Chree, 1913), look for any consistent response in ionospheric measurements made at the Radio Research Station at Slough in the UK. Using historical records, reasonable quantitative estimates of the type and tonnage of explosives for each raid can be made, enabling the raids to be subdivided by size.

## 1.1 Early ionospheric measurements

The Radio Research Station (latterly the Radio and Space Research Station and ultimately the Appleton Laboratory) located at Ditton Park near Slough (Gardiner et al., 1982) conducted routine measurements of the Earth's ionosphere from 1933 to 1996. These measurements continue today at the Rutherford Appleton Laboratory near Chilton, UK. This sequence represents the longest continuous set of ionospheric measurements in the world. The technique used exploits the fact that a transmitted radio pulse is returned from an ionised atmospheric layer when the radio frequency of the pulse, $f$, matches the local plasma frequency. From this, the local electron concentration, $N$, can be determined via the relation $f \approx 9\sqrt{N}$. By transmitting a sequence of radio pulses over a range of radio frequencies, it is therefore possible to construct a height profile of ionospheric electron concentration. Such measurements are most often presented in the form of an ionogram, a plot of "virtual height" (estimated from the time of flight of the signal assuming a vacuum) against radio frequency. From such records the virtual height and the peak frequency (and therefore electron concentra-

tion) returned from each ionospheric layer can be determined and tabulated. While modern digital soundings automatically identify a comprehensive set of such parameters, this was a time-consuming task for the earliest analogue measurements, and so only the peak frequency of the ionospheric F layer (denoted $f_oF2$) was initially routinely scaled. While other parameters were scaled intermittently, $f_oF2$ represents the most comprehensive parameter scaled from these data that exists in a digital form. The original photographic prints of these early ionograms are held by the UK Solar System Data Centre, from which additional information can potentially be gleaned (Davis et al., 2013).

## 1.2 Bombing raids during World War II

Looking for a signature in the UK ionospheric records from Allied bombing campaigns over Europe between 1943 and 1945 may not seem like the most obvious of studies, but there are several reasons as to why a signature from such raids may be detectable over others. While the bombing of London by the Luftwaffe between September 1940 and May 1941 (popularly known as the "London Blitz") would have generated explosions at a closer proximity to the ionospheric measurements being made above Slough, this bombing was more or less continuous, making it difficult to separate the impact of wartime raids from those of natural seasonal variability. In addition, it is well documented that the Luftwaffe did not possess any four-engined long-range bombers (e.g. Beker, 1969). The mainstay of the Luftwaffe was the Heinkel 111, a twin-engined bomber capable of carrying 4400 lb (1667 kg) of bombs. Using external racks, the aircraft could carry 7900 lbs (3600 kg) of bombs, but the external racks blocked the internal bomb bay and significantly impaired the aircraft's performance (Regnat, 2004).

In contrast, the Allied airforces' use of four-engined bombers enabled them to carry much larger bombing loads, including individual high-explosive (HE) bombs of increasing capacity. A regular Avro Lancaster load designed for bombing of heavily industrialised cities (Mason, 1989) consisted of 1 × 4000 lb (1667 kg) amatol-filled ("Cookie"), 3 × 1000 lb (455 kg) minol- or tritonal-filled, impact-fused high-capacity (HC) bomb short-finned, short-delay, tail-armed HE bombs, and up to six additional compartments filled with 4 or 30 lb incendiary bombs. An alternative configuration used for the bombing of factories, railyards and dockyards consisted of 14 × 1000 lb (1667 kg) medium case (MC), general purpose (GP) short-finned HE bombs. The other mainstay of the RAF during this period was the Handley Page Halifax bomber which had a maximum bomb load of 13 000 lbs (5897 kg). Typical loads consisted of six 500 lb, six 1000 lb and two 2000 lb HE bombs or six 500 lb and four 2000 lb HE bombs. Unlike the Lancaster, the Halifax was not able to carry the 4000 lb or larger bombs. A third aircraft used by the RAF in combined or individual raids was the De Havilland Mosquito, which was capable of carrying a single 4000 lb

bomb. The USAAF B-17 "Flying Fortress" bomber was able to carry a bomb load between 4500 lb (2000 kg) and 8000 lb (3600 kg) depending on the range of the mission. This aircraft too was capable of carrying the larger 4000 lb bombs. The use of additional aircraft, such as the Wellington (4500 lb bomb load) and Short Stirling (8000–14 000 lb bomb load) bombers, was gradually phased out during this period of the war.

Amatol, minol and tritonal represent various mixtures of tri-nitro-toluene (TNT) with aluminium or ammonium nitrate. Originally formulated to extend limited supplies of TNT, these mixtures provided similar or even enhanced explosive energy compared with TNT alone (Maienschein, 2002). Torpex was 50 % more powerful than TNT. One metric ton of TNT has an explosive energy of $4.184 \times 10^9$ J.

The "Cookie", used by the RAF, was the first "blockbuster" bomb. The minimum height at which an aircraft could safely drop the 4000 lb "Cookie" without being damaged by the resulting shockwave was considered to be 6000 feet (1800 m). Even so, there are anecdotal accounts of aircraft being damaged despite following this instruction (Nelmes and Jenkins, 2002). Later versions, designed to penetrate and destroy reinforced underground bunkers, were even bigger. These so-called "earthquake" bombs included the 12 000 lb (5450 kg) "Tall Boy" and the 22 000 lb (10 900 kg) "Grand Slam".

While the bombing of London continued during 1943–1945, it was mostly via V1 and V2 rockets which, while individually devastating, did not compare with the explosive energy associated with that of a heavy bombing raid.

## 2 Method

While there was much military activity throughout Europe and beyond during the Second World War, some of the largest bombing raids over Berlin and other European targets took place between 1943 and 1945. The dates of these heavy raids are listed in Table 1, along with an estimate of the combined weight of high explosives used. Where no reference to the total mass of HE bombs could be found for a particular raid, an estimate was made from the recorded number of aircraft, weighted by the mean ratio of HE to total bomb load from raids for which this information was known (0.667). These dates were selected to coincide with the latter part of the war, during which intensive bombing of London was less prevalent and the raids over mainland Europe were more intense (in terms of tonnage of high explosives dropped); the length of time spent bombing was usually much shorter; and more raids occurred during the day (Middlebrook and Everitt, 1985). The times listed in this table mostly came from Berlin air-raid records (Demps, 2004). These were augmented with timings gleaned from various eye-witness accounts (see Appendix A), in particular from the archives of 550 and 410 RAF squadrons whose records have been made

available online. For aircraft records, the times of the first and last recorded bombings were used. For times taken from the times of air-raid alarms, the start and end of these warnings were used. Where no detail about the length of each raid was given, the start and end times are identical. The Berlin air-raid records were recorded in local time, while military records are most likely recorded in GMT. Where known, this time difference has been taken into account, but for some this may introduce an uncertainty of 1 h into the analysis. The start times from these raids were used as "trigger" times in a superposed epoch or Chree analysis (Chree, 1913). This type of analysis is useful in geophysics for detecting a faint signal in a noisy data sequence (e.g. Davis et al., 1997). By calculating the median response of a dataset around these trigger times, any small but repeatable signal is reinforced, while any signal not associated with the trigger events is expected to occur at random and therefore cancel out when averaged. Medians were used in order to minimise the influence of outliers in the analysis.

The ionospheric data used in this study contain a strong seasonal variation introduced by annual and solar cycle variations in ion production and loss. In order that the seasonal distribution did not dominate the signal in any superposed epoch analysis, a 30-day running median was subtracted from these data. Data from the resulting parameter, $\delta foF2$, were then aligned according to the trigger times and combined in the superposed epoch study. Since it is not possible for the ionosphere to be influenced by a given raid prior to its occurrence, the study was further constrained to ensure that each trigger event was aligned with the first subsequent ionospheric data point within a given time window. The length of time between a raid and the subsequent ionospheric measurement, along with the minimum tonnage considered for a "large" raid, are subjective variables in this analysis. In order to address this, the analysis was repeated for a range of time windows, from 0 to $\leq 23$ h, and the size of a bombing raid was defined by the minimum estimated total tonnage of HE used, from $\geq 100$ to $\geq 1000$ t in steps of 100 t.

## 3 Results

The results of a typical superposed epoch analysis using the estimated start time of each bombing raid are shown in Fig. 1. This analysis uses thresholds of $\leq 10$ h of the ionospheric observation for raids using $\geq 700$ t of HE for which there are 14 events. The median ionospheric response is shown as the black line in the top panel. Here the standard error in the median is shown as a grey shaded area around the line, while the dashed and dotted lines represent the 95th and 99th percentiles obtained by repeating the analysis 10 000 times using a similar number of random trigger times drawn from the same epoch for which no major bombing raid has been identified. It can be seen that the ionosphere is significantly weaker (1.9 standard deviations from the mean and around

**Table 1.** Dates, times, locations and estimated tonnage of major bombing raids conducted over Europe between 1943 and 1945. Times for Berlin raids were taken from the duration of air-raid warnings (Demps, 2014). Other times were taken from a variety of eye-witness accounts (see Appendix A). Raids for which the total tonnage of HE bombs was estimated from the type and number of aircraft involved are marked with a $*$. Noon values of $fo$F2 and 30-day running median $fo$F2 values over Slough corresponding to time $= 0$ in the superposed epoch analysis are presented in columns 6 and 7. Of the 152 events considered in this study, there are 29 days for which there is currently no noon $fo$F2 value available.

| Date yyyy/mm/dd | Start time (GMT + 1) | End time (GMT + 1) | HE tonnage | Location | $fo$F2 (MHz) | 30-day median $fo$F2 (MHz) |
|---|---|---|---|---|---|---|
| 1943/01/16 | 19:33 | 21:48 | 150 | Berlin | 5.7 | 5.65 |
| 1943/01/17 | 19:32 | 22:21 | 133 | Berlin | no data | 5.70 |
| 1943/03/01 | 21:39 | 23:50 | 343 | Berlin | 6.4 | 6.20 |
| 1943/03/27 | 22:08 | 00:13 | 429 | Berlin | 5.6 | 5.90 |
| 1943/03/30 | 01:20 | 03:22 | 315 | Berlin | 5.8 | 5.80 |
| 1943/08/23 | 23:41 | 02:35 | 2541* | Berlin | 5.2 | 4.90 |
| 1943/11/18 | 20:11 | 22:23 | 798 | Berlin | 6.1 | 5.80 |
| 1943/11/22 | 19:30 | 21:12 | 1133 | Berlin | 5.6 | 5.70 |
| 1943/11/23 | 19:26 | 21:19 | 710 | Berlin | 5.7 | 5.70 |
| 1943/11/26 | 20:52 | 22:30 | 859 | Berlin | 5.6 | 5.70 |
| 1943/12/02 | 21:31 | 02:04 | 882 | Berlin | 5.6 | 5.70 |
| 1943/12/04 | 04:06 | 04:10 | 2156* | Leipzig | 6.1 | 5.60 |
| 1943/12/16 | 19:24 | 21:04 | 947 | Berlin | no data | 6.15 |
| 1943/12/20 | 19:39 | 19:54 | 2653* | Frankfurt | 6.3 | 6.00 |
| 1943/12/24 | 03:29 | 05:09 | 710 | Berlin | 6.8 | 6.00 |
| 1943/12/29 | 19:23 | 20:23 | 1099 | Berlin | 6.6 | 5.95 |
| 1944/01/02 | 03:13 | 03:18 | 771 | Berlin | 5.8 | 6.00 |
| 1944/01/03 | 01:59 | 03:21 | 658 | Berlin | no data | 6.05 |
| 1944/01/14 | 18:16 | 19:31 | 2092* | Brunswick | 6.1 | 5.80 |
| 1944/01/20 | 18:56 | 20:25 | 1164 | Berlin | 6.4 | 5.85 |
| 1944/01/21 | 23:01 | 23:24 | 2653* | Magdeburg | 5.0 | 5.80 |
| 1944/01/27 | 19:58 | 21:20 | 1067 | Berlin | 4.8 | 5.70 |
| 1944/01/29 | 02:50 | 04:20 | 10866 | Berlin | 5.7 | 5.70 |
| 1944/01/30 | 19:47 | 21:07 | 1069 | Berlin | 5.1 | 5.70 |
| 1944/02/15 | 20:23 | 22:15 | 1203 | Berlin | 4.9 | 5.45 |
| 1944/02/20 | 04:03 | 04:17 | 3367* | Leipzig | 5.5 | 5.40 |
| 1944/02/21 | 04:00 | 04:12 | 2439* | Stuttgart | 5.5 | 5.40 |
| 1944/02/24 | 23:10 | 01:15 | 3002* | Schweinfurt | 5.1 | 5.30 |
| 1944/02/25 | 22:45 | 22:56 | 2000 | Augsburg | 5.8 | 5.30 |
| 1944/03/02 | 03:02 | 03:15 | 2262* | Stuttgart | 5.7 | 5.30 |
| 1944/03/06 | 12:43 | 14:07 | 1196 | Berlin | 4.3 | 5.10 |
| 1944/03/09 | 12:45 | 14:34 | 554 | Berlin | 4.7 | 5.10 |
| 1944/03/15 | 23:18 | 23:28 | 3511* | Stuttgart | 5.4 | 5.10 |
| 1944/03/18 | 22:01 | 22:14 | 3442* | Frankfurt | 5.1 | 4.85 |
| 1944/03/22 | 12:43 | 13:54 | 515 | Berlin | 5.6 | 4.80 |
| 1944/03/22 | 21:50 | 22:05 | 3340* | Frankfurt | 5.6 | 4.80 |
| 1944/03/24 | 21:48 | 23:12 | 1322 | Berlin | 6.0 | 4.80 |
| 1944/03/26 | 22:01 | 22:11 | 2833* | Essen | no data | 4.85 |
| 1944/03/30 | 01:16 | 01:26 | 3253* | Nuremberg | 5.1 | 5.10 |
| 1944/04/09 | 23:55 | 23:55 | 899* | Villeneuve | no data | 4.80 |
| 1944/04/11 | 02:23 | 02:38 | 567* | Aulnoye | 5.2 | 4.80 |
| 1944/04/11 | 22:42 | 22:45 | 1442* | Aachen | 5.2 | 4.80 |
| 1944/04/19 | 02:20 | 02:31 | 1160* | Rouen | 4.8 | 4.80 |
| 1944/04/21 | 02:09 | 02:15 | 1518* | Cologne | no data | 4.80 |
| 1944/04/23 | 01:15 | 01:27 | 2150 | Düsseldorf | 5.1 | 4.80 |
| 1944/04/24 | 23:45 | 01:05 | 2577* | Karlsruhe | 4.5 | 4.80 |
| 1944/04/27 | 01:30 | 01:37 | 1975* | Essen | 4.6 | 4.80 |
| 1944/04/28 | 02:06 | 02:14 | 1234 | Friedrichshafen | 3.5 | 4.80 |
| 1944/04/29 | 11:11 | 12:08 | 709 | Berlin | no data | 4.80 |
| 1944/04/30 | 23:53 | 00:13 | 488* | Maintenon | 5.0 | 4.80 |

| Date yyyy/mm/dd | Start time (GMT + 1) | End time (GMT + 1) | HE tonnage | Location | $fo$F2 (MHz) | 30-day median $fo$F2 (MHz) |
|---|---|---|---|---|---|---|
| 1944/05/02 | 01:05 | 01:16 | 315* | Lyons | 3.9 | 4.80 |
| 1944/05/04 | 00:25 | 00:32 | 1465* | Mailly | 4.5 | 4.80 |
| 1944/05/07 | 10:34 | 11:44 | 1370 | Berlin | no data | 4.80 |
| 1944/05/08 | 10:38 | 11:36 | 858 | Berlin | 4.5 | 4.80 |
| 1944/05/08 | 00:18 | 00:18 | 231* | Rennes | no data | 4.80 |
| 1944/05/10 | 00:11 | 00:12 | 1590* | Mardyck | 5.1 | 4.80 |
| 1944/05/11 | 23:50 | 00:04 | 164* | Hasselt | 5.2 | 4.80 |
| 1944/05/20 | 00:43 | 00:56 | 503* | Orleans | 4.6 | 4.90 |
| 1944/05/24 | 10:30 | 11:34 | 430 | Berlin | no data | 4.90 |
| 1944/05/25 | 02:23 | 02:30 | 1760* | Aachen | 4.8 | 4.90 |
| 1944/05/28 | 02:25 | 02:31 | 687* | Aachen | 5.2 | 4.90 |
| 1944/06/03 | 00:30 | 00:34 | 1053* | Calais | 4.8 | 4.90 |
| 1944/06/05 | 01:14 | 01:18 | 1003* | coastal | 5.2 | 4.95 |
| 1944/06/05 | 23:34 | 23:37 | 5000 | Normandy | 5.2 | 4.95 |
| 1944/06/07 | 01:22 | 02:30 | 3488 | Acheres | no data | 4.95 |
| 1944/06/10 | 03:15 | 03:27 | 1571* | airfields | no data | 4.95 |
| 1944/06/11 | 01:09 | 01:18 | 1299* | railways | 5.0 | 4.95 |
| 1944/06/13 | 00:59 | 01:09 | 1216* | Gelsenkirchen | 4.6 | 4.90 |
| 1944/06/14 | 22:33 | 23:34 | 1230 | Le Havre | 4.6 | 4.80 |
| 1944/06/16 | 01:20 | 01:27 | 1265* | Sterkrade | 4.6 | 4.80 |
| 1944/06/21 | 09:25 | 11:12 | 1220 | Berlin | 4.7 | 4.80 |
| 1944/06/22 | 15:44 | 15:52 | 912* | v-weapons | 5.0 | 4.75 |
| 1944/06/24 | 02:00 | 02:15 | 856* | railways | 4.5 | 4.70 |
| 1944/06/25 | 03:17 | 03:24 | 2929* | v-weapons | no data | 4.70 |
| 1944/06/27 | 03:31 | 03:39 | 2849* | v-weapons | 4.5 | 4.70 |
| 1944/06/30 | 07:55 | 08:06 | 433* | Oisemont | 5.3 | 4.75 |
| 1944/07/02 | 14:14 | 14:21 | 1580* | v-weapons | 4.7 | 4.70 |
| 1944/07/05 | 01:21 | 01:44 | 1189* | railways | no data | 4.70 |
| 1944/07/06 | 01:50 | 02:01 | 647* | Dijon | no data | 4.70 |
| 1944/07/06 | 20:58 | 21:03 | 2139* | v-weapons | no data | 4.70 |
| 1944/07/07 | 21:50 | 22:02 | 2276 | Caen | 4.7 | 4.70 |
| 1944/07/13 | 01:52 | 02:02 | 1594* | railways | 5.1 | 4.70 |
| 1944/07/18 | 05:45 | 05:55 | 6800 | Caen | 4.7 | 4.80 |
| 1944/07/19 | 01:30 | 01:40 | 672* | Scholven | 5.3 | 4.75 |
| 1944/07/20 | 21:00 | 21:01 | 1405* | v-weapons | 4.4 | 4.80 |
| 1944/07/23 | 01:06 | 01:34 | 2583* | Kiel | 4.7 | 4.90 |
| 1944/07/25 | 01:46 | 01:53 | 2540* | Stuttgart | 5.0 | 5.00 |
| 1944/07/26 | 01:57 | 02:11 | 2275* | Stuttgart | 5.0 | 5.00 |
| 1944/07/29 | 01:47 | 02:10 | 2078* | Stuttgart | no data | 4.95 |
| 1944/07/30 | 08:31 | 08:38 | 2753* | Normandy | 4.9 | 4.95 |
| 1944/08/01 | 19:59 | 20:03 | 1405* | Le Havre | no data | 4.95 |
| 1944/08/03 | 14:16 | 20:02 | 4479* | v-weapons | no data | 5.00 |
| 1944/08/04 | 18:01 | 18:08 | 1144* | v-weapons | 5.2 | 5.00 |
| 1944/08/05 | 19:04 | 19:09 | 1286* | oil plants | 5.1 | 5.00 |
| 1944/08/06 | 11:46 | 13:00 | 229 | Berlin | no data | 5.00 |
| 1944/08/08 | 23:19 | 23:24 | 684* | oil plants | 5.0 | 5.00 |
| 1944/08/10 | 12:00 | 12:06 | 416* | Dugny | 5.0 | 5.00 |
| 1944/08/11 | 16:14 | 16:25 | 1816* | railways | no data | 5.00 |
| 1944/08/12 | 15:10 | 15:13 | 287* | u-boats | 4.5 | 5.00 |
| 1944/08/13 | 02:15 | 02:19 | 541* | Falaise | 4.7 | 5.00 |
| 1944/08/14 | 15:29 | 15:39 | 3146* | Normandy | 5.2 | 5.00 |
| 1944/08/15 | 12:02 | 12:04 | 4048* | airfields | 4.7 | 5.00 |
| 1944/08/26 | 01:01 | 01:12 | 1732* | Rüsselsheim | 5.6 | 5.10 |
| 1944/08/26 | 23:10 | 23:14 | 1571* | Keil | 5.6 | 5.10 |
| 1944/08/30 | 02:00 | 02:13 | 1690* | Stettin | 4.6 | 5.20 |

| Date yyyy/mm/dd | Start time (GMT + 1) | End time (GMT + 1) | HE tonnage | Location | foF2 (MHz) | 30-day median foF2 (MHz) |
|---|---|---|---|---|---|---|
| 1944/08/31 | 15:20 | 15:33 | 2364* | v-weapons | 5.5 | 5.20 |
| 1944/09/03 | 17:28 | 17:33 | 2712* | airfields | 5.7 | 5.20 |
| 1944/09/05 | 18:12 | 18:43 | 1339* | Le Havre | no data | 5.20 |
| 1944/09/06 | 09:20 | 09:38 | 1331* | Le Havre | 4.9 | 5.20 |
| 1944/09/08 | 08:45 | 08:45 | 458* | Le Havre | 5.6 | 5.20 |
| 1944/09/10 | 18:55 | 19:30 | 3902* | Le Havre | 5.4 | 5.25 |
| 1944/09/12 | 22:54 | 23:13 | 1589* | Frankfurt | 5.7 | 5.25 |
| 1944/09/16 | 23:45 | 23:47 | 859* | airfields | 5.4 | 5.30 |
| 1944/09/17 | 11:35 | 12:05 | 3000 | Boulogne | 6.3 | 5.40 |
| 1944/09/17 | 18:15 | 18:18 | 487* | Flushing | 6.3 | 5.40 |
| 1944/09/20 | 15:59 | 17:06 | 2534* | Calais | 5.2 | 5.40 |
| 1944/09/23 | 21:19 | 21:30 | 2208* | Neuss | 4.4 | 5.50 |
| 1944/09/26 | 12:21 | 12:27 | 2804* | Calais | 5.1 | 5.70 |
| 1944/09/27 | 10:11 | 10:15 | 1292* | Calais | 5.9 | 5.50 |
| 1944/09/28 | 01:04 | 01:04 | 909 | Kaiserslautern | no data | 5.50 |
| 1944/10/03 | 14:37 | 14:45 | 1065* | Walcheren | 7.3 | 5.70 |
| 1944/10/05 | 22:28 | 22:40 | 2248* | Saabrücken | 6.6 | 5.90 |
| 1944/10/06 | 11:40 | 13:01 | 545 | Berlin | 5.9 | 6.00 |
| 1944/10/07 | 14:00 | 14:28 | 1437* | Emmerich | 6.3 | 6.15 |
| 1944/10/11 | 16:40 | 16:56 | 1172* | Breskens-Flushing | no data | 6.20 |
| 1944/10/14 | 08:08 | 08:57 | 3574 | Duisburg | 5.7 | 6.30 |
| 1944/10/15 | 01:20 | 01:33 | 4040 | Duisburg | no data | 6.30 |
| 1944/10/15 | 19:45 | 19:55 | 2031* | Wilhelmshaven | no data | 6.30 |
| 1944/10/19 | 20:30 | 20:37 | 2389* | Stuttgart | 6.9 | 6.50 |
| 1944/10/23 | 19:30 | 19:53 | 4084 | Essen | 7.6 | 6.60 |
| 1944/10/25 | 15:29 | 15:46 | 4182 | Essen | 6.1 | 6.60 |
| 1944/10/28 | 15:46 | 16:04 | 2940* | Cologne | 7.6 | 6.40 |
| 1944/10/30 | 21:02 | 21:21 | 3431 | Cologne | no data | 6.50 |
| 1944/10/31 | 21:00 | 21:15 | 1972* | Cologne | 6.7 | 6.50 |
| 1944/11/02 | 19:15 | 19:35 | 3957* | Düsseldorf | 7.0 | 6.40 |
| 1944/11/04 | 19:39 | 19:52 | 2947* | Bochum | no data | 6.50 |
| 1944/11/09 | 10:42 | 10:47 | 1093* | Wanne-Eickel | 5.9 | 6.20 |
| 1944/11/11 | 19:02 | 19:06 | 894* | Dortmund | 5.5 | 6.20 |
| 1944/11/16 | 15:30 | 15:34 | 9400 | Düren, Jülich, H | 6.0 | 6.20 |
| 1944/11/18 | 19:03 | 19:03 | 1217* | Wanne-Eickel | 6.6 | 6.20 |
| 1944/11/21 | 19:15 | 19:25 | 1159* | Aschaffenburg | 6.2 | 6.00 |
| 1944/11/27 | 20:01 | 20:08 | 1900 | Freiburg | 6.4 | 6.00 |
| 1944/11/29 | 14:58 | 15:10 | 1249* | Dortmund | 6.6 | 6.00 |
| 1944/12/04 | 19:34 | 19:47 | 2167* | Karlsruhe | 4.9 | 5.80 |
| 1944/12/05 | 10:28 | 11:38 | 1060 | Berlin | 5.5 | 5.90 |
| 1944/12/06 | 20:40 | 20:53 | 2006* | Leuna | 5.5 | 5.90 |
| 1944/12/12 | 19:37 | 19:43 | 2131* | Essen | 6.0 | 5.95 |
| 1944/12/15 | 18:28 | 18:36 | 1386* | Ludwigshafen | 5.2 | 6.00 |
| 1944/12/24 | 18:30 | 18:35 | 412* | Cologne/Nippes | no data | 6.15 |
| 1944/12/29 | 18:58 | 19:06 | 1379* | Scholven Buer | 6.0 | 6.10 |
| 1944/12/31 | 18:46 | 19:00 | 640* | Osterfeld | 5.9 | 6.15 |
| 1945/01/02 | 19:30 | 19:30 | 2166* | Nuremberg | 6.4 | 6.25 |

the 99th percentile level when compared with the range calculated from random trigger events) at time = 0 (the day of the bombing raids). In order to ensure that geomagnetic disturbances did not contribute to the ionospheric response, all events for which Ap exceeded 48 (Kp > 5) were not considered. This is reflected in the low median values in Ap presented in the lower panel.

In order to investigate the sensitivity of this analysis to the arbitrary thresholds applied to the data, the analysis was repeated for a range of thresholds in both the length of time

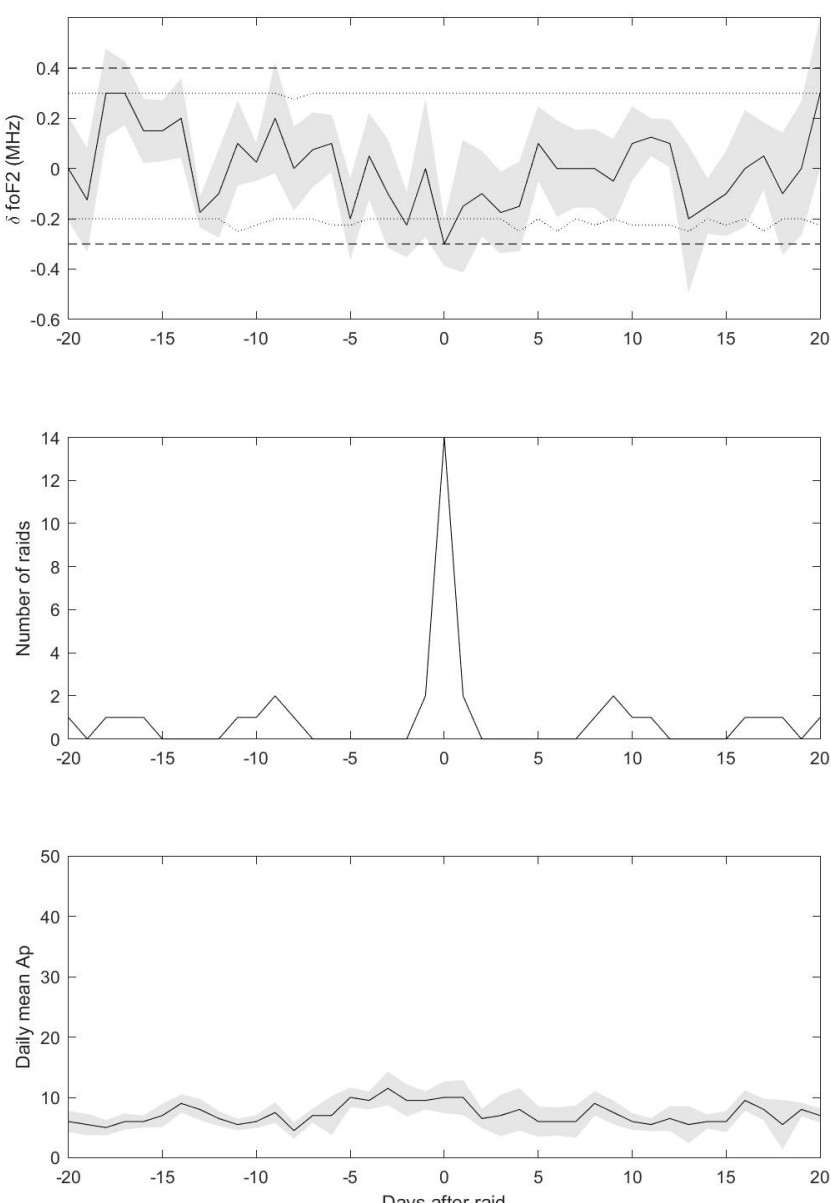

**Figure 1.** A superposed epoch analysis of the ionospheric response to major bombing raids over Europe. The black line in the top panel shows the median response in $\delta fo$F2 ($fo$F2 with a 30-day median subtracted) to 60 bombing raids used as a trigger event in the analysis. The grey shaded area represents the standard error in these median values, while the dashed and dotted lines represent the 95th and 99th percentiles of 10 000 repeated analyses using random control days in which no bombing raids were identified. At time = 0 (within 24 h of the trigger events) $\delta fo$F2 is depleted (1.9 standard deviations from the mean). Arbitrary threshold values were used, corresponding to > 700 t of high explosives per raids occurring within 10 h before the noon ionospheric measurements.

between the raid and the ionospheric measurement (from 0 to 23 h) and the size of a raid as defined by the minimum amount of explosives used (from 100 to 1000 t). The significance of each of these analyses was estimated by calculating the mean and standard deviation of all response times other than at time = 0 and calculating the number of standard deviations the time = 0 measurement was from this mean. The result of these analyses is shown in Fig. 2. The results are plot-

ted on a grid with maximum time between raid and measurement along the $y$ axis and minimum tonnage of the raid along the $x$ axis. The significance of the response at time = 0 for each analysis is shown as a colour contour. As the thresholds used are varied, the number of raids included in each analysis will vary. This is reproduced in a similar grid also presented in Fig. 2, with the colour axis representing the number of events used in each analysis. Two results are clear from this

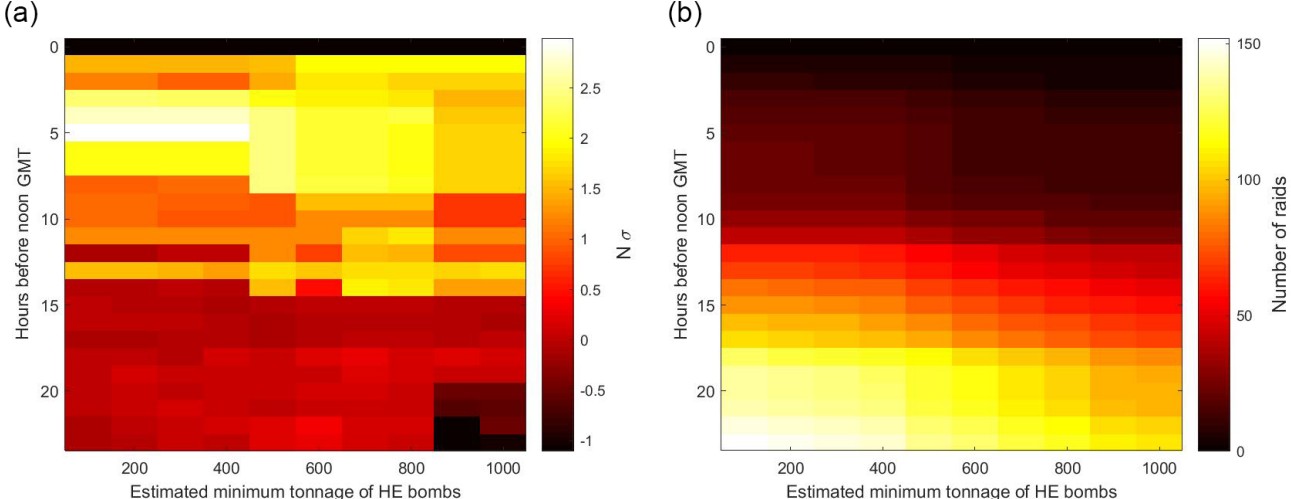

**Figure 2.** Panel **(a)** presents the relative significance of the ionospheric response at time zero (estimated standard deviations from the mean) for a range of thresholds. Events preceding the ionospheric measurements were considered for time windows from 0 to 23 h ahead of the noon ionospheric measurement and the minimum tonnage of HE bombs used in each raid was also varied from 100 to 1000 t. The most significant ionospheric response occurred for events occurring within 5 h before noon, in which a minimum of 300 t of HE bombs were dropped. Panel **(b)** presents the number of events contributing to each analysis. The significance of the response decreases if the threshold is extended beyond 5 h ahead of the ionospheric measurement. This indicates that events occurring at larger time offsets are not contributing to the observed median response. The significance of the result 5 h ahead decreases as the threshold on the minimum tonnage of HE bombs increases. This is likely due to the decreasing number of events contributing to each analysis.

analysis. Firstly a significant ($> 2\sigma$) ionospheric response is obtained for a broad range of trigger thresholds ($3 \leq$ time window $\leq 7$ h & $100 \leq$ minimum tonnage $\leq 800$ t). Secondly, the most significant result ($2.99\sigma$) is seen for a time window of $\leq 5$ h and a minimum tonnage per raid of 300 t. The results of this analysis are presented in Fig. 3, for which there are 20 events. Consideration of the number of events contributing to each study suggests that 14 or more events are required before the random noise is reduced to a level where a significant signal can be detected. The ionospheric responses for the analyses that contain the most events (with thresholds exceeding 15 h) are not significant, indicating that these events do not contribute to the response observed within 5 h of the ionospheric observations.

## 4 Discussion

From the analysis undertaken in this study, no minimum threshold in HE tonnage is resolved. In order to investigate this, data for smaller raids need to be included. In addition to the major raids considered here, there are many more smaller-scale raids involving fewer or smaller aircraft. For example, Mosquito aircraft were used in many hundreds of bombing raids throughout this period (Middlebrook and Everitt, 1985). Given the fact that the current list of major bombing raids used in this study is by no means comprehensive, it is likely that information about many hundreds of smaller raids would need to be included in order to tease

out the signal of such raids from the background ionospheric variability in which further large raids were occurring. As such we consider this beyond the scope of the current study.

For the ionosphere at the altitude of the F2 region ($\sim 200$–$300$ km) above the UK to respond to bombing raids conducted at distances up to 1000 km away, the bombing must have generated pressure waves that were capable of propagating to ionospheric altitudes. A sound wave travelling this distance in the lower atmosphere would arrive within an hour. The speed of sound is temperature-dependent and the temperature decreases with altitude in the troposphere and mesosphere before increasing in the thermosphere. Since the thermosphere represents the most significant fraction of the vertical profile, it is likely that a sound wave propagating vertically as well as horizontally would arrive even sooner. One potential mechanism therefore is of a pressure wave propagating upwards in all directions. At higher altitudes its amplitude increases until it breaks in the upper atmosphere, depositing its energy as heat. A very rough estimate of the anticipated thermospheric temperature rise can be obtained by considering the specific heat capacity of the atmosphere which can be expressed as

$$Q = C_p n \Delta T, \tag{1}$$

where $Q$ is the energy input in joules ($4.184 \times 10^{12}$ for 1000 metric tonnes of TNT), $C_p$ is the molar-specific heat capacity of $N_2$ ($\sim 29.1$ J mol$^{-1}$ K$^{-1}$), $n$ is the number of moles of gas m$^{-3}$ (at ionospheric altitudes, the number density of the atmosphere is $\sim 10^{16}$ m$^{-3}$ or $1.66 \times 10^{-8}$ moles m$^{-3}$) and

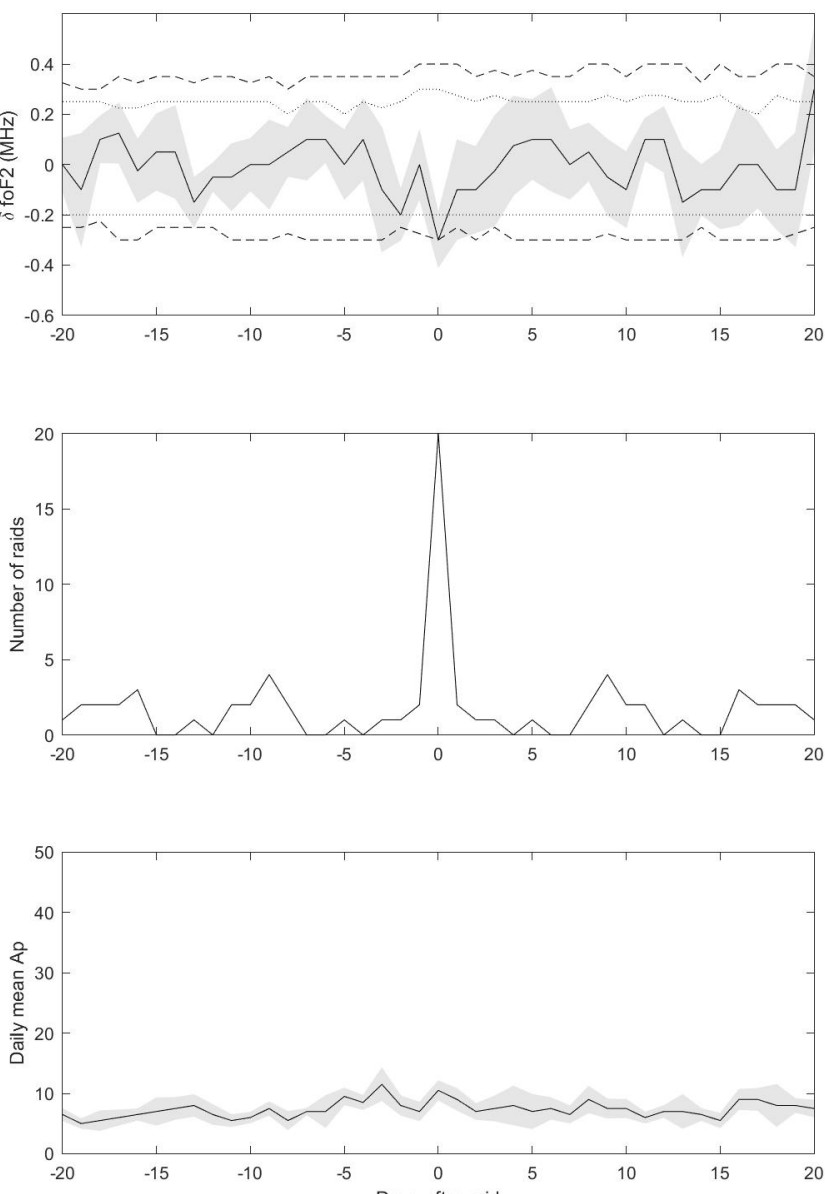

**Figure 3.** The same as Fig. 1, except for the most significant response using 20 bombing raids within threshold values corresponding to > 300 t of high explosives per raids occurring within 5 h before the noon ionospheric measurements. The ionospheric response at time = 0 is around the 99th percentile of 10 000 control studies using random dates on which no bombing raids are catalogued within the study.

$\Delta T$ is the change in temperature (K). Assuming the energy is equally distributed throughout a cylinder of atmosphere 1000 km in radius and 300 km in height, this gives a temperature rise of $\sim 9$ K.

The dominant ion species in the mid-latitude ionospheric F region is $O^+$, whose loss rate is temperature-dependent (Rees, 1989). However, the dominant mechanism by which $O^+$ ions are lost is through reaction with $N_2$ and $O_2$ molecules in the reactions

$$O^+ + N_2 \rightarrow NO^+ + O, \tag{R1}$$
$$O^+ + O_2 \rightarrow O_2^+ + O. \tag{R2}$$

The overall loss rate, $\beta$, for $O^+$ ions can therefore be expressed as

$$\beta = k_1 \cdot [N_2] + k_2 \cdot [O_2], \tag{2}$$

where $[N_2]$ and $[O_2]$ are the concentrations of $N_2$ and $O_2$ molecules respectively and $k_1$ and $k_2$ are the rate coefficients for the two reactions. These rate coefficients are also

temperature-dependent (Rees, 1989). The combined loss rate for $O^+$ ions is therefore dependent on both reaction rates and the concentration of thermospheric species. Müller-Wodarg et al. (1998) modelled the ionospheric and thermospheric response to localised thermospheric cooling ($\leq 40\,K$) during a total solar eclipse. They predicted an $8\,\%$ increase in $fo$F2 ($\sim 0.2\,MHz$) due to the contraction of the atmosphere and an increase in the [O] / [$N_2$] ratio caused, in part, by a reduction in the concentration of $N_2$. It is reasonable to assume that the atmospheric expansion due to energy from localised bombing raids would have an analogous, if opposite, effect on the ionosphere and thermosphere. A rise in the background thermospheric temperature would result in an enhanced loss rate, with the equilibrium between production and loss being established at a lower peak electron concentration. Such equilibrium would be reached within minutes of perturbation, well within the resolution of the ionospheric data. Grandin et al. (2015) studied the impact on $fo$F2 of high-speed streams on Earth. They found that a thermospheric temperature increase of 20–50 K may result in a decrease in $fo$F2 by 0.5–1.0 MHz.

If the bombing resulted in the generation of shock waves or atmospheric gravity waves, their horizontal propagation speed would need to be of the order of $300\,km\,h^{-1}$, while the vertical velocity component would need to be around $100\,km\,h^{-1}$ in order to affect the atmosphere above Slough. There is evidence that turbulence generated in the lower thermosphere by space shuttle launches can propagate 1000 km horizontally within 8 h (Kelley et al., 2009). While this example was specific to the lower thermosphere at altitudes between 100 and 115 km, it nevertheless has a similar time constant to that observed for the ionospheric response to bombing in the current study. Such a mechanism may therefore contribute to the observed effect.

Infrasonic waves generated by explosions are launched preferentially in a vertical direction (e.g. Blanc, 1985). Any impact on the upper atmosphere overhead would then require horizontal transport to move that region over Slough. For the scale sizes involved this would require winds of the order of $300\,km\,h^{-1}$ to blow consistently in a westward direction for the impact to be observed within 3 h, as suggested by the data. For this to happen continuously throughout the epoch being studied is unlikely. Whatever the cause of the ionospheric depletion, the impact appears to last less than 24 h, since only the subsequent noon value is affected.

One metric ton of TNT has an explosive energy of $4.184 \times 10^9\,J$, which is of the same order of energy as a cloud to ground lightning stroke. While a ground-based explosion and a lightning stroke are somewhat different in location and duration, it is not unfeasible that wave energy generated by lightning could also influence the ionosphere in a similar way. Since the occurrence of lightning has distinctive diurnal and seasonal cycles, it is feasible that this mechanism could contribute to the observed seasonal anomaly in

ionospheric F-region electron concentrations (Rishbeth and Müller-Wodarg, 2006).

## 5   Conclusions

We have shown that the influence of major bombing raids over Europe during the latter half of WWII caused a significant depletion of the ionosphere above Slough, UK. This depletion is consistent with an enhanced ionospheric loss rate resulting from thermospheric heating via the dissipation of wave energy in the upper atmosphere. While the list of bombing raids used in this analysis is by no means complete, it is nevertheless sufficient to reduce the background noise in a composite analysis to a level where the ionospheric response to bombing is statistically significant. No lower limit to the minimum mass of explosives required to generate such a response was revealed in this study, though raids using $\geq 100 < 800\,t$ of HE were observed to deplete the ionosphere above Slough if they took place between 3 and 7 h before the ionospheric measurement. It is possible that raids occurring outside of these thresholds could still produce an effect, but the current study contains insufficient data to test this.

One metric ton of TNT has an explosive energy of $4.184 \times 10^9\,J$, which is of the same order of energy as a cloud to ground lightning stroke. Since the occurrence of lightning has distinctive diurnal and seasonal cycles, it is feasible that a similar mechanism could contribute to the observed seasonal anomaly in ionospheric F-region electron concentrations (Rishbeth and Müller-Wodarg, 2006).

The duration of the ionospheric effect appears to be limited to within 24 h. This is currently restricted by the resolution of the digitised data. While hourly $fo$F2 data were digitised from the original analogue ionograms, these data could be re-examined to increase the temporal resolution and investigate the behaviour of other parameters such as the D, E, F1 and sporadic-E layers. The ionospheric sporadic-E layer has already been shown to be enhanced by terrestrial lightning (Davis and Johnson, 2005; Johnson and Davis, 2006; Davis and Lo, 2008). Examining the response of this layer to terrestrial explosions would also provide further information on mechanism(s) involved in this process. The digitisation of these records is therefore highly desirable for a range of different research topics.

*Data availability.* The ionospheric data used in this study are publicly available via the UK Solar System Data Centre (https://www.ukssdc.ac.uk/). The majority of the information about the times, dates and munitions used in the bombing raids was obtained from the published references within this paper. Additional individual sources are cited in Appendix A.

## Appendix A

The information used to determine the times of bombing raids came from a variety of sources. Berlin raids were determined from the information about air-raid alarms detailed in Demps (2014). The timings of most other raids were obtained from the online records of 550 Squadron RAF (http://www.550squadronassociation.org.uk/, last access: 7 September 2018) or 420 Squadron RCAF (http://www.aquatax.ca/snowyowl.html, last access: 7 September 2018). Additional sources are listed below.

### 17 September 1944 – Boulogne

Source: http://www.rafcommands.com/forum/showthread.php?21928-Master-Bomber-required-Boulogne-17th-September-1944&p=128047&styleid$=$3 (last access: 7 September 2018)
"AP 3 – Master Bomber was W/C DM Walbourn, 582 Squadron, H-Hour was 11.35."
"AP 2 –Master Bomber was S/L NS Mingard, 582 Squadron, H-Hour was 12.05."

### 20 September 1944 – Calais

Source: https://www.awm.gov.au/collection/C189567 (last access: 7 September 2018)
"Units: No. 460 Squadron, RAF Bomber Command"
*Accession Number: F02586*
*Place made: France: Nord Pas de Calais, Pas de Calais, Calais*
*Date made: 20 September 1944*
"RAF Bomber Command operational film No. 232. Time 15.59–16.20. Height: 4,300ft. Heavy attack by 646 (27 of 460 Sqn RAAF) Lancasters of Bomber Command on Calais."

### 27/28 September 1944 – Kaiserslautern

Source: http://www.vickersvaliant.com/619-squadron-ops-13—21.html (last access: 7 September 2018)
Extract from the 619 Squadron Operational Record Book (ORB) 27th September 1944: "The primary target, KAISERSLAUTERN, was attacked and bombed from 4,500-ft at 0104 hours." (the mission started on 27 September but bombing occurred in the early hours of 28 September).

### 28 October 1944 – Cologne

Source: http://www.rafcommands.com/forum/showthread.php?7690-Bomber-Losses-28-October-1944 (last access: 7 September 2018)
E.g. "LM182 Primary 20,000ft at 15.46 1/2hrs", "PB56 Primary 19,000ft at 16.04hrs"

### 6 November 1944 – Gelsenkirchen

Source: http://www.lokalkompass.de/gelsenkirchen/politik/6-november-1944-fliegeralarm-in-gelsenkirchen-d24273.html (last access: 7 September 2018)
"Um 13:47 wurde an diesem 6. November Fliegeralarm für Gelsenkirchen ausgelöst. 738 Bomber befanden sich im Anflug auf Gelsenkirchen. Vollalarm. Das Heulen der Sirenen trieb die Bewohner der Stadt in die Bunker und Schutzräume, gegen 14:00 Uhr fielen die ersten Bomben."
– "At 13:47 air raid alarm for Gelsenkirchen was triggered on this 6th of November. 738 bombers were approaching Gelsenkirchen. Full alarm. The howling of the sirens drove the inhabitants of the city into the bunkers and shelters, at 14:00 the first bombs fell."

### 16 November 1944 – Düren, Jülich, Heisburg

Source: http://www.heimatverein-wassenberg.de/images/Wassenberg/archiv/publikationen/sonstige/FrenkenLancaster_PB_137.pdf (last access: 7 September 2018)
"Von immer lauter dröhnendem Motorenlärm alarmiert, befanden sich am 16. November 1944 um 15.30 Uhr alle Besatzungen an ihren Geschützen und gleich der erste Feuerschlag der Batterie Türk gipfelte in einem Volltreffer."
– "Alarmed by the ever louder booming engine noise, all crews were at their guns on 16 November 1944 at 15:30 and the first fire of the Türk battery resulted in a direct hit."

### 13 February 1945 – Dresden

TS1 Source: "Dresden: Tuesday 13th February 1945" by Frederick Taylor, Bloomsburg, 2004.
"At 9.51 p.m. in Dresden the air raid sirens sound, as they have so often during the last five years, and almost always a false alarm"
"The first wave of destruction lasts between fifteen and twenty minutes, the second, two hours later and featuring even more aircraft, last CE1 slightly longer"
Source: http://www.550squadronassociation.org.uk/php-library/mysql-utils/reports/rpt_squadron_operations.php?target$=$dresden (last access: TS2)
E.g. "NN715 Primary at 16,000ft at 01.30hrs", "RA503 Primary at 18,000ft at 01.43hrs"

### 14 February 1945 – Dresden

Source: http://news.bbc.co.uk/onthisday/hi/dates/stories/february/14/newsid_3549000/3549905.stm (last access: TS3)
"The Americans sent 450 B-17 Flying Fortress long-range bombers which arrived at 1230 local time."

### 27 February 1945 – Mainz

Source: http://self.gutenberg.org/article/WHEBN0041209387/Bombing of Mainz in World War II (last access: TS4)

"On 27 February 1945 the RAF flew 435 bombers to attack the city. Between 16:29 and 16:45 hours 1,500 tons of bombs were dropped, hit large parts of the Neustadt. St. Joseph and St. Boniface were destroyed."

### 1 March 1945 – Mannheim

Source: http://www.550squadronassociation.org.uk/php-library/mysql-utils/reports/rpt_squadron_operations.php?target$=$Mannheim (last access: TS5)

E.g.: "26 aircraft with their crews took off at approximately 11.30hrs without incident to participate in a daylight attach on a large concentration of enemy troops reported to be in the vicinity of the Rhine bridgeheads near MANNHEIM"

While no information on time over target is given for this date, from a previous raid over Mannheim on 1 February 1945, the flight time was approximately 4 h, and so the time over target is estimated to be around 15:30 GMT.

### 12 March 1945 – Dortmund

Source: http://www.550squadronassociation.org.uk/php-library/mysql-utils/reports/rpt_squadron_operations.php?target$=$Dortmund (last access: TS6)

"23 aircraft took off without incident at approx. 12.50 hours on what was in almost every respect a repetition of the previous days effort. The only difference, the target was DORTMUND. The colour of the sky markers was changed and the time of the attack about tea time"

Source: https://75nzsquadron.wordpress.com/12345-attack-against-dortmund/ (last access: TS7)

CE2 Bomb load $1 \times 4000$ H.C. $13 \times 500$ ANM.

Primary target – Dortmund

*Tracking error of 0.02* TS8 *large on GH H2S on at 16.50 on run-up to target.*

*Flight*

*Up 13.01 12 March*

*Down 18.16 12 March*

*Total flight time 5 h 15 min*

### 25 April 1945 – Wangerooge

Source: http://www.go2war2.nl/artikel/4413/Air-raid-on-Wangerooge-25-April-1945.htm?page$=$4 (last access: TS9)

"On 16:47 hours, finally an alarm was sounded at Wangerooge by the central communication post on the island that had received messages from transmitting posts on shore. This alarm was directed to the civilian population, that immediately ran for the air raid shelters. A number of them, however, was too late to find refuge. At 16:59 hours, the pathfinding Mosquitoes dropped their coloured markers for the incoming bombers. Less than a minute later the planes came under anti aircraft fire…"

### 25 April 1945 – Berchtesgaden

Source: https://www.awm.gov.au/wartime/61/berchtesgaden (last access: TS10)

"The bombers arrived in two waves at 9 am and 10.30 am."

"Once the target was found, over 1,400 tons of bombs were dropped, including four 12,000 pound Tallboy bombs."

*Author contributions.* CJS collated ionospheric and historical data and carried out the data analysis. PM provided detailed information about the bombing of Berlin and advised on additional historical aspects of the Allied bombing campaign.

*Competing interests.* The authors declare that they have no conflict of interest.

*Acknowledgements.* The authors would like to thank the UK Solar System Data Centre for supplying the ionospheric data used in this study. Detailed information about some of these raids was obtained from various sources (listed in Appendix A), but two particularly useful sources have been the records of 550 Squadron RAF and 420 Squadron RCAF, to whom the authors are extremely grateful for sharing their records. Christopher J. Scott would like to thank Michael Lockwood, Luke A. Barnard, Mathew J. Owens and Clare E. Watt for useful discussions and John D. Davis for his assistance in collating information about the bombing raids. The events that enabled this study represent a period of extreme human tragedy. We dedicate this paper to the aircrew and those on the ground who lost their lives as a result of bombing during WWII.

The topical editor, Dalia Buresova, thanks two anonymous referees for help in evaluating this paper.

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

**Remarks from the language copy-editor**

CE1  "lasts"?

CE2  Should this information relating to the attack on Dortmund be in quotation marks?

**Remarks from the typesetter**

TS1  Please note that we will have to contact the editor if this part of the appendix should be deleted. Please provide a short statement on why this change should be made and I will forward your message to the editor. Note that this comment will be available online after publication.

TS2  Please provide last access date.

TS3  Please provide last access date.

TS4  Please provide last access date.

TS5  Please provide last access date.

TS6  Please provide last access date.

TS7  Please provide last access date.

TS8  Please confirm addition of 0.

TS9  Please provide last access date.

TS10  Please provide last access date.

TS11  Not mentioned in the text.