# Peer review of "The Ionospheric response over the UK to major bombing raids during World War II"

_Annales Geophysicae, 2018_

## Referee Comment (RC1) · Anonymous Referee #1 · 25 Jun 2018

Reviewer's report on

Title: The Ionospheric response over the UK to major bombing raids during World War II
Author(s): Christopher J. Scott and Patrick Major
MS No.: angeo-2018-44

**General comments**

This manuscript presents data of the F2-layer ionospheric variability over Slough, UK observed after 152 major bombing raids over Europe during World War II. Using a superposed epoch analysis, authors found a significant decrease in peak electron concentration (~0.3 MHz decrease in foF2) measured in the noon after the raids. It is suggested that the released explosive energy caused heating the thermosphere, enhancing the temperature dependent loss rate of O+ ions.

This is somewhat unusual paper at a junction of the history and ionospheric physics. I think the material presented here may be interesting not only to ionospheric experts, but also to a broader auditorium.

The paper is clearly written. I may point out only few minor issues to improve or correct.

**Specific comments**

To explain the observed effect, authors suggest the only mechanism, namely (Page 6, lines 24-26):

"The dominant ion species in the mid-latitude ionospheric F-region is O+ whose recombination rate is temperature dependant (Rees, 1989). A rise in the background thermospheric temperature would therefore result in an enhanced loss rate, with the equilibrium between production and loss being established at a lower peak electron concentration, as observed."

The equilibrium between production ($q$) and loss is given by:

$$q = \beta n_e$$

Indeed, in the F peak region, the main ion species is $O^+$, but their recombination rate is very low. So that the F-layer electron loss is dominated by the following two chemical reactions:

$$O^+ + N_2 \rightarrow NO^+ + N$$
$$O^+ + O_2 \rightarrow O_2^+ + O$$

After that, molecular ions ($NO^+$ and $O_2^+$) recombine immediately.

The rates of the reactions ($k_1$ and $k_2$, respectively) depend on the temperature, however the electron loss rate ($\beta$) depends on concentration of the molecules ($N_2$ and $O_2$) as well:

$$\beta = k_1 \cdot [N_2] + k_2 \cdot [O_2]$$

Authors suggest only one mechanism for the foF2 depletion, namely the temperature dependence of $k_1$ and $k_2$, however the thermospheric temperature increase leads also to an increase of the scale height of atmospheric gas $H_s = (k_B T)/mg$ (here $m$ is mass of the molecules). Hence, concentration of $N_2$

and $O_2$ in the F layer peak will increase, which is a second possible reason for increasing the loss rate ($\beta$) and corresponding decrease of the plasma density.

The thermospheric temperature increase may be estimated numerically as

$$\Delta T = Q/C_p n k_B$$

where Cp $\approx$ 3 is the molar heat capacity, $n \approx 10^{10} \text{cm}^{-3}$ is concentration of the atmospheric gas at the F peak, $k_B$ is the Boltzmann constant, and $Q$ is the heat energy per volume. For 1000 metric tons of TNT, assuming the energy was uniformly distributed in the range of 1000 km at height up to 300km, we get: 1000*4.184e9J/(pi*1000km*1000km*300km)/(3*1e10cm-3*1.381e-23J/K) =11K

Grandin et al. [J. Geophys. Res. Space Physics, 2015, doi:10.1002/2015JA021785] studied the ionospheric foF2 decrease caused by the solar wind high speed streams, and have shown that the thermospheric temperature increase by 20-50 K may cause the foF2 decrease of the order of 0.5-1.0 MHz. Hence, energy of the explosions during the raids could potentially cause the 0.3 MHz effect in the foF2, although the above numerical estimates are very rough.

I may mention one more hypothetic mechanism for transport of $N_2$ and $O_2$, namely the turbulence provoked by the shock waves [see e.g., Kelley, et al., (2009), Two-dimensional turbulence, space shuttle plume transport in the thermosphere, and a possible relation to the Great Siberian Impact Event, Geophys. Res. Lett., 36, L14103, doi:10.1029/2009GL038362].

If authors will wish, they may consider these issues in the paper.

For the case if other experts will be interested to make a more comprehensive numerical analysis, I recommend adding in Table 1 two columns showing data of the foF2 for the noon following the raids and the monthly median values.

Finally, I think citation [Kurt Vonnegut (1969), Slaughterhouse-Five, or The Children's Crusade] may be very relevant in the paper.

**Technical comments**

Page 6, line 17: "For the ionosphere (at $\sim$ 250-350 km) above the UK to respond…"
- I suppose authors assume here true height, whereas 250-350 km may be the virtual height measured by the ionosonde (it is typically higher than the true height).

Page 6, line 24: "The dominant ion species in the mid-latitude ionospheric F-region is O+ whose recombination rate is temperature dependant"
- It is correct to say: "…whose loss rate is temperature dependant…"

Page 7, line 1: "Infrasonic waves generated by explosions are launched preferentially in a vertical direction."

- A reference or a more detailed explanation for why it is so will be very relevant here.

---

## Referee Comment (RC2) · Anonymous Referee #2 · 10 Jul 2018

Referee report on the MS "The ionospheric response over the UK to major bombing raids during World War II" by C. J. Scott and P. Major

The subject of the present MS is more or less identical to that provided by Pokhotelov O. A., V. A. Liperovskii, Yu. P. Fomichov, L. N. Roubtsov, O. A. Alimov, Z. S. Sharadze and R. K. Liperovskaya (1991), Ionosphere modifications during military actions in the Persian Gulf war, Dokl. AN SSSR, v. 321, No 6, 1168-1172 published nearly two decades ago. I think that the relevant reference to this MS should be added by the authors. The authors provided quite complex analysis of the ionospheric effects caused by bombing raids during World War II. The results of this study may be published In

Annales Geophys. Discussion.

---

## Author Comment (AC1) · 30 Jul 2018

Many thanks for your detailed consideration of our manuscript. Attached is an archive file detailing our response along with a revised copy of the manuscript.

Please also note the supplement to this comment: https://www.ann-geophys-discuss.net/angeo-2018-44/angeo-2018-44-AC1-supplement.zip

---

## Author Comment (AC2) · 30 Jul 2018

We would like to thank the referee for bringing the reference 'Pokhotelov O. A., V. A. Liperovskii, Yu. P. Fomichov, L. N. Roubtsov, O. A. Alimov, Z. S. Sharadze and R. K. Liperovskaya (1991), Ionosphere modifications during military actions in the Persian Gulf war, Dokl. AN SSSR, v. 321, No 6, 1168-1172' to our attention and we are pleased to include this reference in our manuscript as follows;

"There have been a number of case-studies into the impact of terrestrial explosions on the upper atmosphere (e.g. Rishbeth 1996), most notably the events surrounding the explosion at the Flixborough chemical plant in 1974 (Jones and Spracklen, 1974;

[Figure]

Krasnova et al, 2003) while O. A. Pohotelov et al (1991) considered the impact on the F2 and Es ionospheric layers during a 32 day period of bombing in the Gulf War. In this current paper we make use of historical records to identify large bombing raids over mainland Europe during the Second World War (WWII) and, using a superposed epoch, or composite analysis (Chree, 1913), look for any consistent response in ionospheric measurements made at the Radio Research Station at Slough in the UK. Using historical records, reasonable quantitative estimates of the type and tonnage of explosives for each raid can be made, enabling the raids to be subdivided by size."

Having obtained and translated Pokhotelov et al, we feel that although the nature of the research is indeed similar, there are significant differences between the approach used in the 1991 paper and our current manuscript, most significantly the analysis techniques used, the number of events considered and the quantitative information available regarding the nature and quantity of the explosives used.

We attach a revised version of the manuscript.

Please also note the supplement to this comment:
https://www.ann-geophys-discuss.net/angeo-2018-44/angeo-2018-44-AC2-supplement.pdf

**Supplement:**

[revised manuscript text omitted]